# Glycosylation in Renal Cell Carcinoma: Mechanisms and Clinical Implications

**DOI:** 10.3390/cells11162598

**Published:** 2022-08-20

**Authors:** Xinqing Zhu, Abdullah Al-Danakh, Lin Zhang, Xiaoxin Sun, Yuli Jian, Haotian Wu, Dan Feng, Shujing Wang, Deyong Yang

**Affiliations:** 1Department of Urology, First Affiliated Hospital of Dalian Medical University, Dalian 116021, China; 2Department of Biochemistry and Molecular Biology, Institute of Glycobiology, Dalian Medical University, Dalian 116044, China; 3Department of Surgery, Healinghands Clinic, Dalian 116021, China

**Keywords:** renal cell carcinoma, glycosylation, glycosyltransferase, sialylation, fucosylation, biomarkers

## Abstract

Renal cell carcinoma (RCC) is one of the most prevalent malignant tumors of the urinary system, accounting for around 2% of all cancer diagnoses and deaths worldwide. Clear cell RCC (ccRCC) is the most prevalent and aggressive histology with an unfavorable prognosis and inadequate treatment. Patients’ progression-free survival is considerably improved by surgery; however, 30% of patients develop metastases following surgery. Identifying novel targets and molecular markers for RCC prognostic detection is crucial for more accurate clinical diagnosis and therapy. Glycosylation is a critical post-translational modification (PMT) for cancer cell growth, migration, and invasion, involving the transfer of glycosyl moieties to specific amino acid residues in proteins to form glycosidic bonds through the activity of glycosyltransferases. Most cancers, including RCC, undergo glycosylation changes such as branching, sialylation, and fucosylation. In this review, we discuss the latest findings on the significance of aberrant glycans in the initiation, development, and progression of RCC. The potential biomarkers of altered glycans for the diagnosis and their implications in RCC have been further highlighted.

## 1. Introduction

Renal cell carcinoma (RCC) originates from the epithelium of renal parenchyma tubules, accounting for 80–90% of malignant renal tumors, and is one of the most common malignancies of the urinary system. RCC is responsible for approximately 2% of cancer diagnoses and deaths globally, and the prevalence of RCC in China has been on the rise in recent decades [1,2,3]. RCC is classified into ccRCC, papillary renal cell carcinoma, chromophobe renal cell carcinoma, and collecting duct carcinoma subtypes [4]. Among them, ccRCC is the most common, with a high degree of malignancy, poor patient prognosis, and ineffective chemotherapeutic agents. Surgical treatment has significantly improved the progression-free survival of RCC patients, but 30% of patients experienced metastasis of tumor cells after surgery [5]. In recent years, immunotherapy has made advances in the treatment of ccRCC; however, these advancements are only applicable to a subset of patients [6]. Therefore, identifying prospective targets and acquiring molecular markers for prognostic detection is vital for more precise clinical diagnosis and therapy [7].

Glycans are the most common and diverse biopolymers in nature, attaching to proteins or lipids through glycosylation to generate glycoproteins and glycolipids, respectively, and are one of the four essential components of plant and animal cells. According to the linkage between glycans and the protein core regions, N-glycans, O-glycans, glycosaminoglycans, and glycosphingolipids are the four categories of distinguishable glycans [8,9]. Glycans are involved in biological processes that occur in the tumor, for example, cell-cell signaling and communication, cancer cell metastasis and invasion, angiogenesis of the tumor, and immunological regulation [10].

Glycosylation is one of the most important PTM for proteins that play a crucial part in immunological recognition, cell signaling, and cell-to-cell interaction [11]. Tumor cells exhibit more glycosylation alterations than normal cells [12,13]. Abnormal glycosylation plays a vital role in cancer cell proliferation, invasion, and migration [14,15]. In fact, abnormal glycosylation is associated with differential glycosyl-transferase expression and glycosidases, producing specific glycosylated proteins associated with tumors in cancer cells [16]. Glycosyltransferases are responsible for transferring monosaccharides (usually Nucleoside Diphosphate) from active donors to sugar, protein, lipid, and nucleic acid molecules to complete the glycosylation reaction; glycosidases, on the other hand, are responsible for catalyzing the hydrolysis of glycosidic bonds in complex sugars. Alterations in glycosylation regulate the formation and evolution of cancer, provide a significant contribution as a biomarker, and offer a variety of therapeutic intervention targets [17]. Currently, fucosylation and sialylation are important glycosylation modifications in tumorigenesis and cancer development. Studying the corresponding molecular regulatory mechanism of glycosyltransferases and glycosidases will help us understand their roles in the occurrence and development of RCC. Recently, extensive advancement has been achieved in studying abnormal glycosylation in RCC, which provides a reference for diagnosing and treating RCC. This article reviews the current research progress on the role of abnormal glycans in RCC.

## 2. N-Glycan

The N-glycans share a pentasaccharide core region, which may be further characterized as high mannose type, heterogeneous, or complicated, and altered by the terminal structures GlcNAc, Gal, and sialic acid. N-glycans modify proteins’ asparagine residues (Asn) Figure 1 [18,19]. Evidence shows that abnormal N-glycosylation of proteins is involved in signal transduction, cell adhesion, and tumor occurrence and progression [20,21,22,23]. N-glycosylation is catalyzed by a variety of glycosyltransferases and hydrolyzed by a variety of glycosidases, and the abnormal expression of glycosyltransferases and glycosidases results in alterations in the sugar chain structure. Hai, Zhang J et al. showed that β4GalT-II upregulation in non-metastatic ccRCC patients positively correlated with tumor size, Fuhrman grading, transverse muscle differentiation, tumor necrosis, and recurrence and influenced tumor development and dissemination by affecting the production of cell surface glycoprotein complex-type N-glycans [24]. Li Haoming et al. discovered that the ccRCC TNM staging and overall survival correlated with the downregulation of MAN1C1. By increasing the degree of N-glycosylation, MAN1C1 inhibits cell growth, migration, and invasiveness, and enhances apoptotic [25]. Another study proved that MAN1C1 inhibits epithelial to mesenchymal transition (EMT) through increasing E- cadherin expression [26]. These results suggest that further study of the function and mechanism of N-glycosylation associated with RCC will facilitate our search for potential biomarkers that could provide new diagnostic options and targets for RCC treatment. Fucosylation, sialylation, β1,6-branch GlcNAc, and bisecting GlcNAc are the most common aberrant N-glycosylation kinds associated with the onset and progression of RCC (Figure 2A), as explained below.

### 2.1. Fucosylation

Fucose is a kind of 6-deoxyhexose that is found in several glycoproteins and glycolipids produced by mammals. Fucose can connect to the terminal portion of the N-, O-lipid-linkage oligosaccharide chain, modify the N-glycans core protein, or directly attach to the threonine or serine portion of the protein [27]. According to fucose’s position, it can be divided into terminal fucose and core fucose. Fucose is involved in forming blood grouping H and Lewis antigens, leukocyte selectin that mediates exosmosis or housing, pathogen-host interaction, and signaling pathways alteration [28,29]. There is abnormal fucosylation in various cancers, for example, RCC, breast tumor, and non-small cell lung malignancy, which are crucial for tumor growth, invasion, metastasis, immune escape, and drug sensitivity [30,31,32]. Therefore, targeting the abnormal fucosylation in the tumor may become a new strategy for treating renal cancer and other malignant tumors.

A variety of fucosyltransferases (FUT1-FUT11) are involved in fucosylation. FUT 1 and FUT 2 take part in the α 1,2 fucose formation, while FUT 3-7 and FUT9-11 participate in the formation of α 1,3 and α 1,4 fucose [29]. FUT8 is involved in α1,6 fucose synthesis and the modification of core fucosylation that adds fucose to the GlcNAc part of the asparagine linkage. FUT3 was found to be a predictor of ccRCC patients’ overall and recurrence-free survival. FUT3 gene shows overexpression in gastric tumors, colon cancer, and RCC [33,34]. Due to its essential involvement in the biosynthesis of Lewis antigen (Le) and sialic acid Lewis antigen (SLe), the FUT3 gene is also known as the Lewis gene [29,35]. FUT3-mediated partial fucosylation causes the EMT, augments tumor cell-macrophage communication, and enhances malignant transformation and immune evasion in ccRCC [36]. FUT11 catalyzes the transfer of α-1,3-fucose from GDP fucose to N-and O-linked glycans, free oligosaccharides, lipids, or directly to proteins [37]. El plutona zodro et al. analyzed differential genes between renal cell cancer and healthy tissues and identified their upregulation in the microarray dataset. In addition, qPCR found 14 upregulated genes in 32 ccRCC tumor samples. The data indicated that FUT11 was the glycosyltransferase with the highest differential expression. Previous research has found that, when compared to healthy tissue, ccRCC had a greater expression of fucose [38]. El plutona zodro et al. analyzed that the upregulation of FUT11 in ccRCC may be related to the development and course of ccRCC [39]. Therefore, further experiments are needed to explore the mechanism of FUT11 in ccRCC, which will provide a theoretical basis for our further understanding of rockweed glycosylation during the development of ccRCC.

### 2.2. Sialylation

Sialic acid (SA) is a type of acidic monosaccharide consisting of 9-carbon that has a negative charge, a natural carbohydrate widely found in the biological system. Sialylation modification is one of the main forms of glycosylation modification, which refers to linking the sialic acid moiety of the sialic acid donor CMP-Neu5Ac to the ends of glycoproteins (N-glycans, O-glycans), and glycolipids via (α 2,3, α 2,6, α 2,8) bonds in the presence of sialyltransferases (STs) [40,41].

Sialylation plays a critical part in various biological processes, including adhesion to the cell, recognition of antigens, and transduction of signals [42,43]. Tumor cell expression of sialic acid is associated with cancer behaviors such as the proliferation of cells, migration, and invasion [44]. It is well established that alteration of ST activity is one of the fundamental processes behind abnormal sialylation of glycans in cancer. A study finds abnormally high expression of ST3Gal-I is associated with decreased overall survival in RCC [45]. It was also shown that miR-193a-3p and miR-224 directly inhibit the expression of ST3Gal-IV through the PI3K/Akt pathway to promote RCC’s proliferation, migration, and invasion ability [46]. The results of Hai-Ou Liu et al. exhibited that the ST6Gal-I expression in RCC was significantly correlated with the poor prognosis of histopathological phenotype [47]. Compared with ST6Gal-I low expression patients, the prognosis of patients with high expression of ST6Gal-I is significantly worse. It is currently hypothesized that ST6Gal-I suppresses apoptosis by increasing α2,6- sialylation of FASR to prevent activation of the death-inducing signaling complex (DISC) and binding of the Fas-associated adapter molecule (FADD) to the FASR death domain [48,49]. Pan Y et al. found that HOTAIR acts as a competitive endogenous RNA (ceRNA) that regulate ST8SIA-IV expression by binding to miR-124, thereby promoting proliferation and metastasis of RCC [50]. In RCC development, the HOTAIR/miR-124/ST8SIA-IV axis is a potential therapeutic target for RCC, which provides new insights into RCC treatment.

Furthermore, sialic acid is a component of the oligosaccharide determinant cluster of the Lewis type of sialic acid, ST3Gal-III, ST3Gal-IV, and ST3Gal-VI catalyzes the synthesis of sialylated Lewis antigens [51]. Borzym-Kluczyk et al. demonstrated that the expression of sialylated Lewis antigens is increased in the tissues of RCC patients, which predicts a poor prognosis and a predisposition for lymph node metastases [52]. In clinical trials, an increasing number of studies concentrating on sialylation have proven substantial effects on the treatment of cancers in recent years. These results suggest that focusing on STs may be a novel treatment strategy for RCC patients.

### 2.3. β1-6 Branching GlcNAc

N-acetylglucosamine aminotransferase V(GnT-V)) has been shown to catalyze the synthesis of β1-6 branching GlcNAc on N-glycans [53]. Numerous studies have shown that β 1-6 branching GlcNAc facilitates the expansion of the ectodomain and modifies the glycan structure of glycoproteins cell surface; for example, calmodulin, cell surface growth factor, and integrin, that share in the formation and progression of a broad spectrum of cancers, oncogenic growth signaling, tumor migration, and invasion processes [54,55,56,57,58]. GnT-V has been stated to promote the synthesis of β1-6GlcNAc branch on β1 integrin, which then inhibits cisplatin-induced apoptosis or α5β1 integrin aggregation and promotes migration of cells in cervical squamous cell carcinoma and fibrosarcoma [22]. In RCC, the internalization and recirculation of β1 integrins can regulate the malignancy of cancer cells. In addition, previous studies have shown that GnT-V also plays a vital part in the immunological response. GnT-V can control T cell activities and autoimmunity through the alteration of oligosaccharides on T cell receptors [59].

Moreover, GnT-V-mediated N-glycosylation controls Th1 cytokine production and macrophage phagocytosis in vivo negatively [60]. Liu Y et al. suggested that ccRCC TNM and metastasis correlated positively with the expression of the GnT-V, and these patients had a bad prognosis; even after multivariate Cox regression tests, its expression was considered an independent poor predictive factor affecting the survival and recurrence of ccRCC [61]. Therefore, GnT-V could be identified as targeted therapy, and inhibiting GnT-V might improve the ccRCC antitumor effect.

### 2.4. Bisecting GlcNAc

The bisecting GlcNAc is catalyzed by β1,4-N-acetylglucosamine transferase III (GnT-III), which connects GlcNAC to the mannose residues of N-glycans with β1,4-glycosidic bonds. The glycosylation modification of bisecting GlcNAc is involved in biological processes such as intracellular signaling, nerve development and regeneration, cell adhesion, tumor growth, infiltration, and metastasis [62,63]. Previous study has shown that bisecting GlcNAC can block the hypoxia-induced epithelial to mesenchymal transition of breast cancer cells [64]. A study based on RCC tissues showed that the activities of GnT-III are significantly reduced, resulting in a decrease in bisecting GlcNAc [61]. These findings suggest that the alteration in bisecting GlcNAc could be a marker of RCC pathogenesis. Therefore, studying the structural changes of bisecting GlcNAc and the aberrant expression of GnT-III will aid in advancing our understanding of its role in RCC development and provide a theoretical basis for the search for potential biomarkers that can be used in the diagnosis and treatment of RCC.

## 3. O-Glycans

O-glycans modify threonine or serine residues of proteins and are the second most common glycan structure after N-glycans. Depending on the combination of added sugars, four common O-glycan core structures (core1-core4) are found in mammals (Figure 2B). Core1 synthase first transforms the basic GalNAc-1-Ser/Thr into Gal1-3GalNAc-1-Ser/Thr (core1). The core1 was then transformed into core2 by β-1,6-N-acetylglucosaminyltransferase (core2 β1-6GlcNAc transferase or C2GnT). Core2 O-glycan has been shown to be highly expressed in cancer tissue, whereas core3 and core4 O-glycans have been reported to be downregulated in malignant tissue [65,66,67]. Dystroglycan (DG) is a cell surface laminin receptor that connects the extracellular matrix with the actin cytoskeleton. The primary O-mannosylation of DG requires the combined activity of (O-mannosyltransferase 1,2 (POMT1,2) and isoprene synthetase domain protein (ISPD) [68]. After that, DG needs LARGE, LARGE2, and β3 GnT-I to phosphorylate its O-mannose for recognition through a large number of O-glycan chain extensions [69]. Michael R et al. revealed that loss of DG glycosylation in ccRCC patients was associated with high mortality, with the most significant difference in the *GYLTL1B* gene encoding LARGE2, which was reduced by nearly 80% in ccRCC patients compared to normal controls [70]. In addition, loss of DG glycosylation affects ccRCC invasion and proliferation. This suggests that the *GYLTL1B* gene mediates the decrease in DG glycosylation in ccRCC by decreasing the expression of LARGE2. Thus, *GYLTL1B* might be a potential targeted therapy for ccRCC, while DG could be a potential prognostic biomarker for ccRCC. In conclusion, future studies on the structure and function of O-glycans will provide novel insights into the role of glycosylation in RCC patients and their survival.

### 3.1. O-GalNAc

Mucin-type O-glycosylation is catalyzed using polypeptide N- acetylgalactosaminyltransferases (GALNTs) that transfer GalNAc from the UDP donor to certain Thr or Ser residues of the acceptor to generate O-GalNAc glycans (mucins) [71]. The simplest mucin-type O-glycan is GalNAc α1-Ser/Thr (Tn antigen), while sialic acid Tn antigens (STn) is a truncated O-glycan that has α-2,6 sialic acid that is linked to GalNAc α-O-Ser/Thr [72,73]. STn expression in more than 80% of tumors and changes in its levels are associated with cancer activities such as cell-to-cell adhesion, cellular migration, cell invasion, and immune regulation [74,75]. In recent years, ST6GalNAc-I has been widely recognized for its role in tumor activity by regulating Tn and STn antigens [76]. Qi Bai et al. detected ST6GaNAc-1 high expression in tumors by tissue microarray and immunohistochemical staining and showed that ST6GalNAC-I affected the patient’s overall survival in non-metastatic ccRCC [77]. Mucins contain a large amount of O-glycosylation, and abnormal expression or glycosylation of mucins can alter the biological behaviors of tumor cells, such as growth, differentiation, adhesion, and invasion [78,79]. It was shown that MUC1 expression was upregulated in ccRCC and was found to be a possible promoter of renal cancer invasion and metastasis during epithelial-mesenchymal transition. Furthermore, combining HIF-1α with the MUC1 promoter under hypoxic conditions resulted in the overexpression of MUC1, thus promoting the invasive and migratory properties of ccRCC [80]. Tian Niu, Xu Zhiying et al. detected high expression of MUC3A and MUC13 in patients with non-metastatic ccRCC by tissue microarray and immunohistochemistry and found that the expression of MUC3A and MUC13 was positively correlated with Fuhrman classification [81,82]. Although the mechanisms of MUC3A and MUC13 in the carcinogenesis and progression of ccRCC have not been investigated, it has been shown that the expression of MUC3A might be controlled through the PKC signaling pathway, which regulates the metastatic invasiveness of tumor [83]; MUC3A overexpression is linked with the reduction of tumor suppressor gene p53 and activation of critical oncogenes HER2, PAK1, ERK, and Akt [84]. This suggests that MUC3A and MUC 13 may become immunotherapeutic targets for ccRCC, and further studies on the biological mechanisms involved in the progression of ccRCC by MUC3A and MUC 13 are needed, that can be promising strategies in the treatment of ccRCC patients.

### 3.2. O-GlcNAc

Another post-translational alteration is known as O-linked N-acetylglucosamine (O-GlcNAc or O-GlcNAcylation) [85]. The synthesis of O-GlcNAc is activated by O-GlcNAc transferase (OGT), which transfers GlcNAc from UDP-GlcNAc into serine or threonine of protein substrates [86]. Abnormal O-GlcNAc regulation shares in the formation and development of various illnesses. O-GlcNAc primarily has a biological function in carcinogenesis through O-GlcNAcylating proteins such as p53 and β-catenin [87,88]. Recently, it has been observed that O-GlcNAcylation plays a role in a variety of human malignant tumors, including prostate cancer, colon cancer, lung cancer, breast cancer, etc. [89,90]. Wang L et al. discovered that O-GlcNAcylation levels and OGT expression in renal cancer cell lines and tissue samples were considerably higher than in normal controls. Simultaneously, increased OGT expression was linked to RCC Fuhrman classification and poor prognosis [91]. The effect of O-GlcNAcylation on apoptosis and cell cycle distribution in renal cell carcinoma was studied using flow cytometry. The findings indicated that OGT overexpression might increase RCC cell proliferation by blocking apoptosis and accelerating the cell cycle, suggesting that O-GlcNAcylation and high OGT expression are oncogenic factors in the genesis of renal carcinoma. Therefore, OGT may become a target for future RCC therapy, which provides new insights into the selective regulation of O-glycosylation on complex glycosylation in ccRCC. Table 1 summarizes the abnormal glycosyltransferases related to ccRCC and the mechanism of action in ccRCC. 

## 4. Proteoglycans

Proteoglycans (PGs) are macromolecular sugar complexes formed by 1 or more glycosaminoglycans (CAGs) covalently bonded to the core protein of varying lengths. CAGs are long linear heterogeneous polysaccharides composed of repeating disaccharide units that differ in their basic composition, connection, acetylation, N-sulfation, and O-sulfation of sugar; these disaccharide units are galactose, galactosamine, N-glyogalactosamine-4-sulfuric acid, and galacturonic acid. It is mainly divided into two categories: guanidine sulfates (hyaluronic acid) and sulfates (chondroitin sulfate, heparin sulfate, keratin sulfate, and dermatan sulfate) [95]. GAGs play an important function in maintaining cellular integrity and tissue normal structure. Electrostatic interactions between positively charged amino acids and negatively charged glucuronic and sulfate groups in proteins are the primary mechanism by which these molecules attach to their respective protein targets. It is now known that they interact with cytokines, chemokines, growth factors, and enzymes and play a crucial part in coagulation, growth, infection, inflammation, tumor formation, and metastasis [96,97,98]. F. Gatto and colleagues studied a novel systems biology method in order to evaluate the significance of GAG biosynthesis regulation in ccRCC [99]. The results revealed that GAG levels were markedly high in ccRCC patients than in normal tissues, and changes in GAG composition, sulfation, and chain length were observed. It can be confirmed that changes in GAG are important metabolic events during ccRCC transformation. Therefore, GAG was shown to have a high sensitivity diagnostic and prognostic value for the surgical treatment of ccRCC [100]. Lucas et al. estimated the expression of glycosaminoglycan profiles and heparinase in the early stage RCC patient’s tissue. The results revealed that heparinase and chondroitin sulfate expression was significantly higher in early RCC tissues than in non-tumor tissues. Heparanase produces heparan sulfate oligosaccharides by degrading the heparan sulfate chains in proteoglycans, thereby promoting proliferation, angiogenesis, and migratory ability of tumor cells [101]. Thus, chondroitin sulfate and heparinase expression could be used as transitional molecules to evaluate non-neoplastic and renal cell carcinomas, providing new targets for treating RCC. 

Hyaluronic acid (HA) is a ubiquitous macromolecular glycosaminoglycan synthesized by three hyaluronan synthases, HAS1, HAS2, and HAS3. In recent years, an increasing number of researchers have focused on the relationship between tumorigenesis and HAS1 and showed the role of HAS1 in the growth of RCC and other malignant tumors [92]. The results of Mao-Kun Sun et al. showed that miR-125a could inhibit RCC cell migration and invasion by targeting STAT3 to regulate the expression of HAS1 [93]. In summary, the abnormal expression of PGs in tumor cells might be used as a marker for cancer progression and patient. Improving the understanding of PGs metabolic regulation and PGs involvement in tumors provides a new avenue for targeted therapy of the tumor microenvironment.

## 5. Glycosphingolipid

Glycosphingolipids (GSLs) are polysaccharides that bind a lipid core ceramide (Cer) and more than 300 complex compounds, primarily cerebrosides and gangliosides [102]. The diversity of polysaccharide structure on GSLs is determined using a series of proteins shared in polysaccharide synthesis, including glycosyltransferase (GTs), glycosidases, and enzymes involved in the synthesis of polysaccharide precursors and nucleotide sugar transporters. Cer can be galactosylated in the endoplasmic reticulum to produce glucosylceramide (GalCer), sphingomyelin in the golgi matrix, or glucosylated to produce glucosinoceramide (GlcCer) [103]. GSLs are a vital part of the cell membrane that plays a vital part in molecular signal transmission, regulation of membrane protein function, and cell adhesion [104]. Several studies have shown that glycosphingolipids may play a role in the pathophysiology of commonly clinical diseases, including those associated with abnormal kidney growth [105]. Aerts JM et al. have shown that Fabry’s disease is caused by impaired GSL metabolism, which accumulates deacylceramide and deacylsphingosine in lysosomes and other membranous cavities, leading to the progressive loss of renal function [106]. Previous studies have shown that monosialylgalactosylglobulin (MSGG) affects the biosynthesis of disialylgalactosylglobulin (DSGG) by decreasing the activity of ST6GalNAc-VI, which, in turn, increases the ability of RCC to metastasize [107]. Satoh m et al. revealed that in addition to promoting the development and metastasis of renal cancer, gangliosides produced by renal cancer cells could indirectly promote tumor growth by inhibiting the function of immune cells [108]. Some reports have indicated that gangliosides block many steps of the cellular immune response, including antigen processing, presentation, T cell proliferation, and production of cytokines, for example, IFN-c and IL4 [109,110]. Consistent with ganglioside synthesis, several vital enzymes regulating ganglioside synthesis are abnormally expressed in renal cell carcinoma, including β-1,4-GalNAc transferase (β-1,4-GalNAc-T), ST3Gal-II, and the plasma membrane-associated sialic acidase NEU 3. The study by Cristina Tringali showed that the mRNA expression level of plasma membrane sialidase NEU3 in RCC is significantly higher than that of non-tumor tissues adjacent to cancer. NEU 3 enhances the invasiveness and migration of RCC by down-regulating the content of ganglioside GD1a. NEU3 inhibits the endocytosis of β1 integrin in RCC, promotes integrin recycling, increases integrin levels in the cell plasma membrane, and activates the EGFR and FAK/Akt EGFR and FAK/Akt signaling pathways [94]. Therefore, NEU3 is expected to be a novel molecular target for the specific treatment of RCC patients.

## 6. Glycans as Therapeutic Targets in Renal Cancer

In a personalized medicine scenario, medicines directed against certain tumor entities are chosen depending on the characteristics of the patient’s cancer molecules. Markers that show how resistant or sensitive a patient is to a certain treatment plan have been proposed to optimize therapy. Clinical outcomes will continue to improve as new and improved therapy choices, and stratifying biomarkers become available. Glycosylation has been demonstrated to be a significant structural component not just in biomarkers but also in therapy. It has the potential to enrich the tools and techniques now available for precision medicine. This section describes recent breakthroughs in glycan-based medications, including immunotherapy, antibody-based therapy, glycan mimetics, vaccine development, glycoconjugate drug development, glycan-based targeted nanotherapies, and clinically useful glycosylation inhibitors. It has been demonstrated that tumor cells that display glycosylation patterns have an effect on the outcome of RTK targeting. For example, it has been demonstrated that overexpression of the ST6GAL1 transferase, which is linked to poor prediction in many cancer types, confers resistance to trastuzumab (anti-HER2 antibody) and gefitinib (EGFR inhibitors) -mediated programmed cell death of tumor cells [111,112]. Consistent with these findings, gefitinib resistant lung cancer lines show an overall increase in sialylation than sensitive lines. Another significant glycan modification linked with cancer development is the increase in core fucosylation, which FUT8 mediates [112]. In liver and ovarian malignancies, core fucosylation’s role in acquiring cancer characteristics has been established. In this regard, interesting findings have been produced using small compounds which disrupt the N-glycosylation process partially, hence causing cancer cell death that is addicted to RTKs and RTKs inhibitors [113]. On the other hand, some research suggests that targeting glycosylation in cancer immunotherapy may be beneficial. For example, targeting the PD-L1 glycoform has emerged as an approach for optimizing immunotherapy, and attempts are underway to incorporate PD-L1 glycan epitopes to stratify marker panels for anti-PD-L1/PD-1 [112,114]. Additionally, sialylation reduction or interference with immunosuppressive receptors that identify sialic acids has emerged as a promising method for cancer immunotherapy, owing to the critical function of cancer cell sialylation in the establishment of an immune-suppressive milieu. In this context, inhibiting sialylation in vivo using drugs that mimicked sialic acid led to an increase in the amount of tumor cell death caused by CD8+ T cells. Additionally, a fraction of cancer-infiltrating T cells that express Siglec-9 seems to be an essential part of the immunosuppressed cancer microenvironment, which suggests that these cells might be utilized as a target for immunotherapy in order to boost T cell activation [112]. Figure 3 summarizes the altered glycans associated with oncogenic pathways, their role in RCC development, proliferation, progression, invasion, and metastasis, and the therapeutic target based on these glycans.

### 6.1. Glycan-Based Vaccines

The glycosylation mechanism goes through a unique change during the development of the tumor, which leads to an increase in the number of branching highly glycosylated structures. This change is typically accompanied by differential glycosylation of mucin core proteins, including MUC1 and MUC16. It has been determined that tumor-associated carbohydrate antigens, often known as TACA, should be the major focus of efforts to create tumor vaccines. Covalently attaching carbohydrates to immunologically active proteins significantly increases their immunogenicity. Given that the linker molecule used to connect the carbohydrate to the protein can affect the immunological characteristics, it is critical to use immunologically inactive linkers when synthesizing conjugates [115]. According to the TACA frequency utilized in manufacturing conjugates, glycoconjugates vaccines might be classified as mono-epitopic, mono-epitopic cluster, or multi-epitopic vaccines comprising one type of TACA, cluster of one type of TACA, or a combination of different types of TACA. Mono-epitopic vaccination has been more widely studied and tested in clinical testing than other types [116]. In the realm of glycan-based-tumor therapeutics, the invention of the Globo H hexasaccharide vaccine conjugated to keyhole limpet hemocyanin for patients with prostate cancer is regarded as a watershed event [117]. The synthetic Globo H-keyhole limpet hemocyanin (KLH) conjugates and the immunologic adjuvant QS-21 were shown to be a well-tolerated immunization for breast cancer patients in phase 1 clinical study [17]. The Thomsen-Friedenreich (TF)-KLH-QS21 vaccine demonstrated potential effectiveness with higher IgM and IgG antibody titers in a phase 1 clinical study in individuals with recurrent prostate cancer [17]. Scientists have studied the prospect of producing multi-epitopic cancer vaccines that target multiple cell populations in light of the link between various TACA and tumor progression. As a consequence, the same research team revised the vaccination to contain five carbohydrate antigens associated with the advancement of prostate and breast cancer, Globo-H, TF, GM2, TF, STn, and Tn [118]. A three-component vaccination based on a TLR2 agonist, a T helper epitope, and a tumor-associated glycopeptide (TACA) elicited high titers of IgG antibodies specific for TACA in a similar experiment by another group [17,115]. Sialyl-Tn antigen, a carbohydrate linked with MUC1 that is not expressed in healthy tissues, was identified as a promising candidate for inclusion in the vaccine. The breast cancer vaccine created by covalently binding synthetic STn antigen to KLH protein has advanced to phase 3 clinical trials [119]. Clinical studies have shown that none of the vaccines have been effective in lowering illness progression or overall survival rates, which is unfortunate. Vaccines made from carbohydrate-protein conjugates, which exhibit heterogeneity and imprecision as a consequence of varying physical, chemical, and immunological characteristics, have been shown in studies to have certain intrinsic drawbacks. This might influence the repeatability of the immune response. As a result, scientists have focused their efforts on developing synthetic conjugate vaccines with well-defined structures and no immunosuppressive effects in order to increase vaccine immunogenicity. It was discovered that a glycopeptide with a cluster of three Le(y)-serine epitopes was more effective than one containing a single Le(y) epitope. Numerous vaccines have been created with synthetic carbohydrates with minor structural characteristics and beneficial T cell-dependent immune responses. It has been shown that a synthetic vaccine made up of the lipopeptide Pam3Cys, Tn antigen, and YAF peptide is efficient in inducing an immune response in the treatment of cancer [120]. TLR2 agonist, T-helper epitope, and TCGA were covalently linked in another research, producing an antibody with strong anti-breast cancer IgG titers. To make an anti-cancer vaccine with higher immunogenicity, scientists used click-chemistry for solid-phase synthesis of MUC1 glycopeptide and Pam(3) Cys lipopeptide [121]. There were many other vaccinations that were extremely effective in killing cancer cells and generating an immunogenic response, such as synthetic glycan vaccines [118]. Tumor size decreased after immunization with many of these synthetic vaccines in vivo. However, further research is required to evaluate the clinical significance of this finding.

### 6.2. Glycoconjugate Drugs

On the way to improving clinical results, chemotherapeutic drugs face major challenges, including greater toxicity and poor selectivity. Various attempts have been made to avoid this, including the glycoconjugate of medications. The year 1995 marked the first production of sugar conjugated compounds by Pohl et al. [122]. The glycoconjugate medicine (glufosfamide) showed a 4.5-fold decrease in toxicity in rats. Later, in phase II clinical study, twenty patients with resistant solid tumors proved the effectiveness of glufosfamide [123]. There seem to be a number of more intriguing instances in which carbohydrate-modified drugs are utilized to enhance the efficacy of chemotherapy treatments. For example, the cardiotoxicity of Adriamycin is significantly reduced when it is conjugated with 2-amino-2-deoxy-d-glucose and succinic acid. This also reduces the drug’s ability to inhibit the activity of several other drugs. Numerous newly developed anti-cancer medications, such as paclitaxel, azomycin, ketoprofen, cadalene, docetaxel, and chlorambucil, have been shown to have promising anti-cancer properties if they are combined with monosaccharides [17].

### 6.3. Glycosylation Inhibitors

Novel molecular glycosylation inhibitors are being investigated for potential as anti-cancer drugs. Most glycosylation antagonists function by blocking intracellular functions or precursor processing. The majority of inhibitors are made up of very tiny compounds that a wide variety of cell types may readily absorb. Providing a promising new target for the development of drugs to treat a wide range of diseases caused by aberrant glycosylation. There are now several glycosylation antagonists in clinical trials for cancer therapy. Acute Myeloid Leukemia patients who used Uproleselan (GMI-1271) in conjunction with an E-Selectin antagonist had a high percentage of complete remission and excellent survival outcomes [124]. Pembrolizumab and the galectin inhibitor GR-MD-02 have been used in combination as a therapy for melanoma, non-small cell lung cancer, and head and neck squamous cell carcinoma. Another galectin blocker, GMCT-01, is being tested in combination with the chemotherapeutic drug 5-fluorouracil in a phase 2 clinical study for malignancy. Breast, lung, head and neck, and prostate cancers are all being studied in phase 1 clinical trials using GM-CT-01 in conjunction with fluorouracil. Renal, urinary bladder, gastric, breast colorectal, squamous cell carcinoma, head and neck, and non-small cell lung cancers are all being tested in phase 1 clinical study using SGN-2FF [17].

### 6.4. Glycan-Based Nanotherapy

There is a growing consensus that glycan-based polymers, such as hydrogels, micelles, and nanoparticles, are preferable candidates for use as delivery vehicles for chemotherapeutic agents. According to the findings of Puranik and colleagues, nanoscale hydrogels exhibit excellent physicochemical features and are safe enough for oral delivery of hydrophobic chemotherapeutic medicines [125]. Recently published techniques have developed carbohydrate-based nanomaterials that are ideally suited for chemotherapeutics controlled release. According to research, using polymeric nanogels in conjunction with cisplatin and gemcitabine as a treatment for pancreatic cancer significantly improves the effectiveness of the treatment and its systemic delivery. Researchers discovered that placing cisplatin inside a polymeric nanogel that had cross-linked ionic cores led to an increase in cisplatin accumulation at the site of the tumor as well as an improvement in the drug’s safety profile. For targeted administration, the STn antigen, which is abundantly expressed in cancer cells, was conjugated to nanogel-encapsulated cisplatin using the TKH2 monoclonal antibody. When gemcitabine is administered in conjunction with TKH2-functionalized cisplatin-loaded nanogels, tumor development may be effectively reduced in vitro and in vivo [126]. One other team has created a biocompatible nanoparticle that selectively transports 5-fluorouracil and Paclitaxel to SLeA-expressing gastric cancer cells while avoiding healthy tissue. Investigators have used polysaccharides including chitosan, chondroitin sulfate, alginate, and hyaluronic acid (HA) in order to create nanoparticles due to their exceptional physicochemical features, allowing for more regulated and focused pharmacological action. Carbohydrate-based polymers enhance the solubility of insoluble or poorly soluble medicines, boost the effectiveness of therapy, and reduce unwanted effects through changing treatments’ absorption, distribution, and metabolism. As a result, carbohydrate-based polymers are widely regarded as a potential material for the production of effective nanocarriers for anti-cancer therapy regimens [17]. Overall, aberrant protein glycosylation in cancer offers researchers more options for identifying potential biomarkers.

## 7. Conclusions and Future Perspectives

In this review, we described the glycosylation modifications associated with RCC and gave insights into exploiting aberrant glycosylation, sialylation, fucosylation, and glycan branching in an effort to identify alternative therapeutic strategies for RCC therapy. The study of glycosylation alterations in RCC can contribute to a deeper knowledge of the disease’s mechanisms and the identification of potential biomarkers and therapeutic targets. Decades of research indicate that glycosylation alterations in RCC are important and observable. Combining glycomics and glycoproteomics with genomes, transcriptomics, proteomics, and metabolomics has allowed the detection of altered glycoproteins and glycolipids and specific glycan structures linked with cancer in a small number of samples. Even though glycan changes in human cancer cells were found decades ago, our understanding of how glycosylation affects the growth of tumors has sped up in recent years.

## Figures and Tables

**Figure 1 cells-11-02598-f001:**
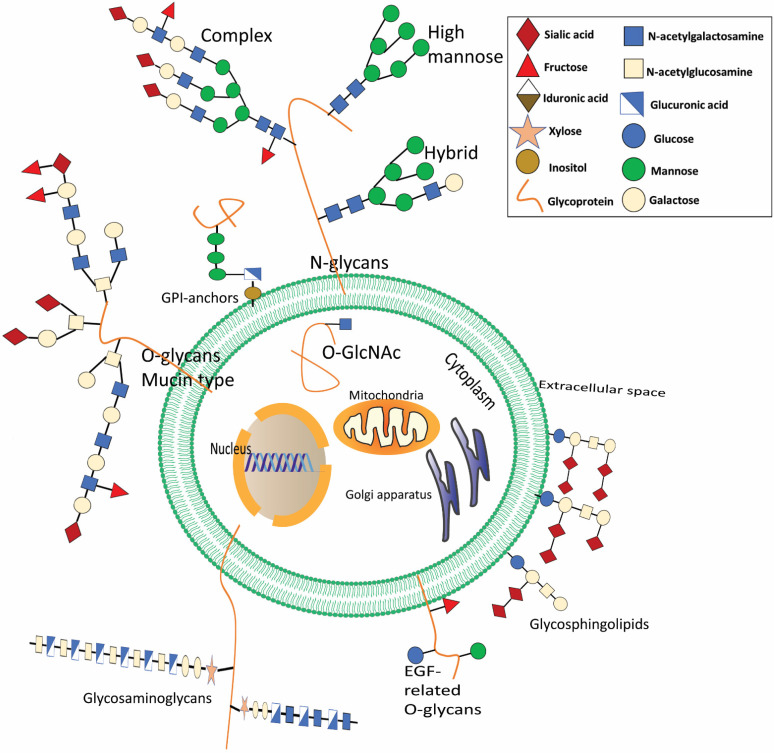
The most common glycoconjugates present in mammals are depicted. Glycans are present in a wide range of biomolecules. Glycosphingolipids are essential components of the cell’s external membrane. A reversible sequence of structures, called ceramide-linked glycans, are modified by terminal sialic acid. Glycosylation occurs when an N- or O-linked saccharide is attached covalently to Asp or Ser/Thr8 in the polypeptide backbone. N-Glycans all have the same basic structure (N-acetylgalactosamine), but they may be further subdivided into high mannose, hybrid, and complex forms, and their terminal structures can further be modified. When O-linked with Ser/Thr, it will develop mucin-type O, which is often seen in membrane-associated or secreted glycoproteins. Some glycoproteins are also linked to phosphatidylinositol in the plasma membrane’s outer leaflet, called glycosylphosphatidylinositol (GPI) linked proteins. Less common O-glycans may alter epidermal growth factor (EGF)-like repeats (O-fucose, O-glucose, and O-linked N-acetylglucosamine). Glycosaminoglycans are longitudinal co-polymers of acidic disaccharide repeating units found as hyaluronic acid or connected to proteoglycans in the extracellular matrix. Glycosaminoglycans may be sulfated at the N-site (NS) or at multiple O-sites, including 2-O-sulfation (2S), 4-O-sulfation (4S), and 6-O-sulfation (6S). Glycosphingolipids are ceramide-linked glycans that dominate the exterior cellular membranes. In addition, O-GlcNAcylation adds O-linked N-acetylglucosamine to various cytoplasmic and nuclear proteins.

**Figure 2 cells-11-02598-f002:**
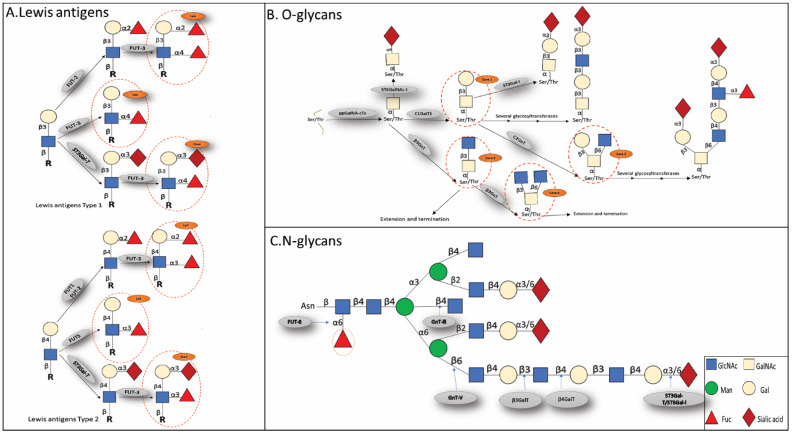
The figure shows some of the important glycan moieties that will be talked about in this study. (**A**) Terminal Lewis and sialylated Lewis; (**B**) O-linked and (**C**) N-linked glycans. Lewis a (Lea), Lewis b (Leb), and sialyl Lewis a (SLea) are type 1 Lewis antigens. Lex, Ley, and SLey are type 2 Lewis antigens. Truncated O-glycans such as T, sialyl Tn(STn), or Tn are often present in cancer cells. Additionally, terminal glycosyl epitopes, such as sialyl Lewis(SLe) epitopes, α 2,3-, or α 2,6-linked sialic acid to N—acetyllactosamine(SLN), and fucosylated Lewis (Le) structures may be found at the ends of the glycan chains. The N-glycan structures of both bisecting N-acetylglucosamine and β 1,6-branching N-acetylglucosamine compete with one another. In cancer, the balance is often tipped in favor of branching. N-Glycans can also be changed by adding fucose to their center. Gangliosides are sialylated glycosphingolipids. Sialylated glycosphingolipids are known as gangliosides. GM3, GD3, or GD2 are often overproduced in the abnormal form in cancer. The key enzymes in charge of adding certain sugar residues are also shown in boxes. GalNAc-T1, GalNAc-T2, GalNAc-T3, GalNAc-T4, GalNAc-T5, and GalNAc-T6 are among the 20 enzymes that make up the family of polypeptide N-acetylgalactosamine transferases (ppGalNA-Ts), sialyltransferases (like-galactoside-2,6 sialyltransferases I (ST6Gal I). FUT8, which helps add “core” 1,6 Fuc to N-glycans; FUT1 and FUT2, that add fucose (Fuc) for 1,2 linkage to galactose (Gal); FUTs that help add Fuc in 1,3 linkage to 2,3 sialylated type 2 chain (FUT3, FUT4, FUT5).

**Figure 3 cells-11-02598-f003:**
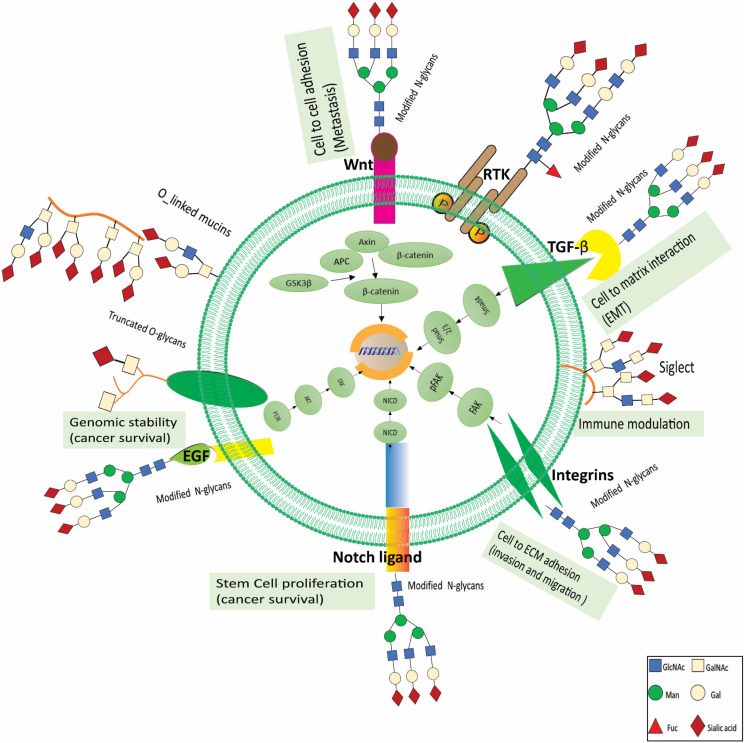
Glycans’ role in RCC growth and progression. Glycans serve critical roles in the pathological stages of tumor formation and progression. Glycans disrupt cell-cell adhesion during cancer cell dissociation and invasion. Epithelial cadherin (E-cadherin) is changed with 1,6-N-acetylglucosamine (1,6GlcNAc)-branched N-glycan structures when the activity of N-acetylglucosaminyltransferase V (GnT V) is elevated. This decreases cell attachment and enhances tumor cell invasion. Such highly branched patterns may be enlarged, and the terminal structures that are 2,6-sialylated inhibit the attachment of tumor cells. Protein stability and the suppression of tumor development may be attributed to the existence of E-cadherin N-glycans that include bisecting GlcNAc structures. This reaction is catalyzed by GnT-III. In addition, abnormal O-glycosylation is associated with tumor cell invasion. One example of this is the production of sialyl Tn (STn), which can be caused by either overexpression of -N-acetylgalactosamine (-GalNAc)-2,6 sialyltransferase I (ST6GalNAc I) or mutations in C1GALTT1-specific chaperone 1 (C1GALT1C1). The formation and proliferation of tumors are characterized by changes in glycosylation of key growth factor receptors, which modifies the receptors’ functions and the signals they send. Ganglioside expression in the membrane of cancer cells may also interfere with signal transmission, setting off a wide array of cellular pathways that foster the growth and development of tumors. Alterations in O-GlcNAcylation have been connected to the development of cancer as well. During the migration of tumor cells, integrins display various glycosylation patterns in O-linked and N-linked glycans. Terminal sialylation disrupts the connections between cells and their extracellular matrix (ECM), which results in an invading and migrating phenotype. The abnormal glycosylation of the vascular endothelial growth factor receptor (VEGFR) changes the way the receptor interacts with galectins and has been associated with the development of tumor angiogenesis. Cancer-associated carbohydrate determinants sialyl lewis x (SLex) and sialyl lewis ea (SLea) function as ligands for adhesion receptors expressed in activated endothelium cells (E-selectin), platelets (P-selectin), and leukocytes (L-selectin), which facilitates cancer cell adherence and metastasis. RTKs are triggered by changes in receptor glycosylation, gangliosides, and glycosaminoglycan expression, resulting in enhanced cancer cell motility, invasion, and proliferation. Fuc stands for fucose; Gal stands for galactose; GlcA stands for glucuronic acid; Man stands for mannose; RTK is for receptor tyrosine kinase; Xyl stands for xylose.

**Table 1 cells-11-02598-t001:** Glycosylation and its functions in renal cell carcinoma.

Glycan Components	Involved Glycosyltransferases		Impact on Renal Cell Carcinoma	References
β1,4-galactose	β4GalT-II	-	Promoting	[24]
α1,2-mannose	MAN1C1	E-cadherin	Suppression	[25,26]
α1,3-fucose/LeX	FUT3	-	Promoting	[33,34,35]
α1,3-fucose	FUT11	-	-	[37,38,39]
α2,3-galactose	ST3Gal-I	-	Promoting	[45]
α2,3-galactose	ST3Gal-IV	miR-193a-3p, miR-224	Suppression	[46]
2,6-galactose	ST6Gal-I	DISC, FADD, FASR	Promoting	[47,48,49]
α2,8-galactose	ST84sia4	HOTAIR, miR-124	Promoting	[50]
sLeX	ST3Gal-III, ST3Gal-IV, ST3Gal-VI	-	Promoting	[51,52]
β1,6-branching GlcNAc	GnT-V	β1 integrin	Promoting	[54,55,56,57,58,59,60,61]
Bisecting GlcNAc	GnT-III	-	Suppression	[62,63]
core2 O-glycan	C2GnT	-	Promoting	[65,66,67]
STn antigen	ST6GalNAc-I	-	Promoting	[76,77]
O-GalNAc	MUC1 MUC13 MUC3A	HIF-1α - p53, HER2, PAK1, ERK	Promoting Promoting Promoting	[80] [81,82] [83,84]
O-GlcNAc	OGT	-	Promoting	[91]
Hyaluronic acid (HA)	HAS1, HAS2, HAS3	miR-125a, STAT3, β1 integrin, EGFR	Suppression	[92,93]
Ganglioside	NEU3	GD1a	Promoting	[94]

## Data Availability

Not applicable.

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
