# Peer review of "Glycosylation in Renal Cell Carcinoma: Mechanisms and Clinical Implications"

_cells, 2022, doi:10.3390/cells11162598_

Round 1

Reviewer 1 Report

I have gone through the research manuscript entitled: “Glycosylation in renal cell carcinoma: mechanisms and clinical implications”. This is a carefully done study and the findings are of considerable interest, originality and thus merit for publication to the Cells. The language of the manuscript is written well and expressed clearly. Describing the many glycan expressing pattern including N-glycans, O-glycans, proteoglycans, glycosphingolipids will be the target for therapeutic targets in cancer.

However I found some points to correct to improve this manuscript shown as below.

-Schematic diagram of glycan should be make more carefully (Figure1, 2, 3).

Figure 1: For instance, the shape of cartoon of monosaccharides in glycans are different. Some of the mannose looks circle but ellipse in N-glycans and GPI anchors. Shape of monosaccharides (Squares) of Glycosaminoglycans could improve.

Figure 2:

1) The size and shape of monosaccharides are not unified even it is same monosaccharides. For example, sialic acid in O-glycan. The balance of size among monosaccharides are weird in N-glycan.

2) Whole schematic diagram of B. (O-glycan) and C. (N-glycans) are not inserted in Figure 2.

3) It says “A. Lewis antigens”, “C. N-glycans”. Should “B.O-glycan” be “O-glycans”?

4) Is there necessity to separate the rounded square between Lewis antigens type 1 and type 2? Both of the glycans belongs to Lewis antigen so it can integrate to single rounded square. And some words and dotted circle appeared around Lewis antigens type 2. It should erase them.

5) Add the indication of square, circle… of monosaccharides as you did in Figure 1. It is easier to understand the structure.

Figure 3:

Same comments as Figure 1 and 2. Unified the size and shape of monosaccharides.

-Small correction.

1) Page 7, title of 3.2. “O-GlcNac” should be “O-GlcNAc”. Some are described as “O-GlcNAC”. It should be integrated.

2) Page14, line 4 from the head of Section 6.3 Inhibitors of glycosylation, “providing a promising ….”, it should be “Providing a promising…”

Author Response

Dear Professors:

On behalf of my co-authors, thank you very much for giving us an opportunity to revise our manuscript, we appreciate the editor and reviewer very much for worthy comments and response to the manuscript entitled “Glycosylation in renal cell carcinoma: mechanisms and clinical implications”. Actually, we have studied the reviewer’s comments carefully and working on your comments to improve the revised paper. Please find the revised version.

Manuscript revised comprehensively by a professional native speaker. you may see a revised manuscript.

We would like to express our great appreciation to you and the reviewers for their comments on our paper. Looking forward to hearing from you.

Thank you and best regards.

Yours sincerely,

Abdullah Al-Danakh

Deyong Yang

Reviewer #1:

I have gone through the research manuscript entitled: “Glycosylation in renal cell carcinoma: mechanisms and clinical implications”. This is a carefully done study and the findings are of considerable interest, originality and thus merit for publication to the Cells. The language of the manuscript is written well and expressed clearly. Describing the many glycan expressing pattern including N-glycans, O-glycans, proteoglycans, glycosphingolipids will be the target for therapeutic targets in cancer.

However, I found some points to correct to improve this manuscript shown as below.

Comment 1:
-Schematic diagram of glycan should be make more carefully (Figure1, 2, 3).

Reply to comment1:

Ok, thank you for your comment, it’s fixed now you may see new updated Figures 1, 2, 3.

Comment 2:
Figure 1: For instance, the shape of cartoon of monosaccharides in glycans are different. Some of the mannose looks circle but ellipse in N-glycans and GPI anchors. Shape of monosaccharides (Squares) of Glycosaminoglycans could improve.

Reply to comment 2:

Thanks, as you recommend, the shapes of all monosaccharides are now same in size and uploaded in better quality you may see Figure 1

Comment 3:

 Figure 2:

The size and shape of monosaccharides are not unified even it is same monosaccharides. For example, sialic acid in O-glycan. The balance of size among monosaccharides are weird in N-glycan.

Reply to comment 3:

We are sorry for such a mistake. It is fixed, you may see Figure 2

Comment 4:

 Figure 2:

2) Whole schematic diagram of B. (O-glycan) and C. (N-glycans) are not inserted in Figure 2.

Reply to comment 4:

It was designing problem , it is fixed now and whole schematic diagram of B. (O-glycan) and C. (N-glycans) are now  inserted ,you may see figure 2

Comment 5:

 Figure 2:

3) It says “A. Lewis antigens”, “C. N-glycans”. Should “B.O-glycan” be “O-glycans”?

Reply to comment 5:

Thanks for the valuable notice, we fix it and we write “O-glycans” instead of O-glycan, you may see figure 2  .

Comment 6:

 Figure 2:

Is there necessity to separate the rounded square between Lewis antigens type 1 and type 2? Both of the glycans belongs to Lewis antigen so it can integrate to single rounded square. And some words and dotted circle appeared around Lewis antigens type 2. It should erase them.

Reply to comment 6:

Yes you are right ,we integrate them  and we delete dots and words that appear around Lewis antigens type 2 you may see Figure 2.

Comment 7:

 Figure 2:

 Add the indication of square, circle… of monosaccharides as you did in Figure 1. It is easier to understand the structure.

Reply to comment 7:

It is a very valuable recommendation; it is added, you may see figure 2. If you have any further consideration, we are happy to hear from you.

Comment 8:

Figure 3:

Same comments as Figure 1 and 2. Unified the size and shape of monosaccharides.

Reply to comment 8:

Ok, we fixed all sizes and shape and make them unified and better quality, and more understandable; you may see Figure 3

Comment 9:

 Page 7, title of 3.2. “O-GlcNac” should be “O-GlcNAc”. Some are described as “O-GlcNAC”. It should be integrated

 Reply to comment 9:

It is fixed now; you may see line 263

Comment 10:

Page14, line 4 from the head of Section 6.3 Inhibitors of glycosylation, “providing a promising ….”, it should be “Providing a promising…”

 Reply to comment 10:

Thanks for all your comments. It is fixed; you may see line 470

Warm regards 

Reviewer 2 Report

The present manuscript is a review on Glycosylation in renal cell carcinoma: mechanisms and clinical implications. As a reader, I had the expectation that the glycan structure would be the central point manifesting different facades of renal carcinoma. Nevertheless, the review is a bit sketchy. With such an important topic, I would suggest little more in-depth writing is expected. glycans can play very crucial role as biomarker, disease-progression marker, resistance, treatment outcome etc. But Which type of sugar and/or glycosidic linkage and where getting altered to have a particular disease manifestation and what is the mechanism?...are questions of this minute.  

Author Response

Dear Professors:

On behalf of my co-authors, thank you very much for giving us an opportunity to revise our manuscript, we appreciate the editor and reviewer very much for worthy comments and response to the manuscript entitled “Glycosylation in renal cell carcinoma: mechanisms and clinical implications”. Actually, we have studied the reviewer’s comments carefully and working on your comments to improve the revised paper. Please find the revised version.

Manuscript revised comprehensively by a professional native speaker. you may see a revised manuscript.

We would like to express our great appreciation to you and the reviewers for their comments on our paper. Looking forward to hearing from you.

Thank you and best regards.

Yours sincerely,

Abdullah Al-Danakh

Deyong Yang

Reviewer #2:

Comment 1

The present manuscript is a review on Glycosylation in renal cell carcinoma: mechanisms and clinical implications. As a reader, I had the expectation that the glycan structure would be the central point manifesting different facades of renal carcinoma. Nevertheless, the review is a bit sketchy. With such an important topic, I would suggest little more in-depth writing is expected. glycans can play very crucial role as biomarker, disease-progression marker, resistance, treatment outcome etc. But Which type of sugar and/or glycosidic linkage and where getting altered to have a particular disease manifestation and what is the mechanism?...are questions of this minute.  

Reply to comment 1

Thank you very much, you are correct, the glycan structure plays a crucial role in renal cancer, and we have revised the manuscript to make it more understandable to researchers and rephrase some sentences. In addition, we emphasize the importance of glycans and modification toy may see the revised manuscript. Finally, we have updated the abstract and conclusion following your recommendation; see lines (12-24) and (512-522). If you have more considerations, we are excited to hear them.

Comment 2

English language and style are fine/minor spell check required

Reply to comment 13

 Manuscript revised comprehensively by a professional native speaker. you may see a revised manuscript.

Best regards